# Protein–Protein Interactions Efficiently Modeled by Residue Cluster Classes

**DOI:** 10.3390/ijms21134787

**Published:** 2020-07-06

**Authors:** Albros Hermes Poot Velez, Fernando Fontove, Gabriel Del Rio

**Affiliations:** 1Department of biochemistry and structural biology, Instituto de fisiologia celular, UNAM Mexico City 04510, Mexico; albros.poot@gmail.com; 2C3 consensus, Leon Guanajuato 37266, Mexico; fernando.fontove@c3consensus.com

**Keywords:** residue cluster class, protein–protein interaction, machine learning

## Abstract

Predicting protein–protein interactions (PPI) represents an important challenge in structural bioinformatics. Current computational methods display different degrees of accuracy when predicting these interactions. Different factors were proposed to help improve these predictions, including choosing the proper descriptors of proteins to represent these interactions, among others. In the current work, we provide a representative protein structure that is amenable to PPI classification using machine learning approaches, referred to as residue cluster classes. Through sampling and optimization, we identified the best algorithm–parameter pair to classify PPI from more than 360 different training sets. We tested these classifiers against PPI datasets that were not included in the training set but shared sequence similarity with proteins in the training set to reproduce the situation of most proteins sharing sequence similarity with others. We identified a model with almost no PPI error (96–99% of correctly classified instances) and showed that residue cluster classes of protein pairs displayed a distinct pattern between positive and negative protein interactions. Our results indicated that residue cluster classes are structural features relevant to model PPI and provide a novel tool to mathematically model the protein structure/function relationship.

## 1. Introduction

Proteins perform many vital functions in living organisms, with most depending on interactions with other molecules. Among these interactions, protein–protein interactions (PPI) are involved in maintaining cellular structure, regulating protein function, facilitating cellular transport, and ultimately encoding for the scaffold where most, if not all, cellular events take place [1,2,3]. Hence, identifying these PPI represent an important effort to characterize the molecular mechanisms at play in different living organisms.

To facilitate this effort, different computational approaches were described over the past years. Of particular interest to this work were those approaches based on machine learning (ML) techniques [4,5,6]. In ML, two global approaches are recognized, namely, supervised and unsupervised learning. In the latter, clusters of elements are identified and guided mainly according to the distance separating positive and negative PPI, while in supervised learning the algorithms thrive in identifying the frontiers of known clusters [7]. ML works on numerical representation of proteins; hence physical and chemical descriptors were developed to represent proteins [8,9,10] and used to predict PPI [11,12]. Other descriptors, such as proteins sequence composition [13,14], genomic data [15,16], and protein three-dimensional (3D) structures [17,18], among others [19,20], were described to represent proteins to predict PPI.

We previously reported a compact, 26-dimensional representation of protein structure based on residue cluster classes (RCCs) [21], which are obtained from the maximal cliques observed in the contact map derived from the protein 3D structure, taking into account the sequence localization of amino acid residues. RCCs are sets of maximal cliques that are grouped by size and classified according to the sequence proximity of the included residues (see Figure 1). RCCs represent the denser packing areas of a protein (every residue in these clusters is in contact with the rest). We showed that RCCs present a pattern that is recognizable by any heuristic ML approach and consequently provided a learnable representation for protein structure classification. Indeed, we showed that RCCs improved upon the state-of-the-art methods aimed to look for structural neighbors and structural classification [21]. In the present work, we aimed to test if protein structure is key to protein interactions through the use of RCCs to classify PPI.

## 2. Results

Experimentally determined positive examples of PPI were obtained from the three-dimensional interacting domains (3DID) database [22] and negative ones from the Negatome database [23] (see Methods); only proteins in these sets with 3D structures reported in the public repository of protein structures, Protein Data Bank (PDB), were included in this study. These sets rendered a database with 171,142 pairs of positive PPI and 692 pairs of proteins known to not interact (negative PPI) to use for training (see Methods), representing 99.6% and 0.4% of the training set, respectively. For every protein in these sets, we used 12 different representations of RCCs that were generated using 12 different distance criteria (see Methods). We also created two different sets of RCCs either including or not including the atoms of the sidechains of residues. Hence, every protein was represented in 24 different RCCs. Each of these representations was trained to model PPI either by adding every pair of RCCs or by producing the concatenation set (see Methods); in this way, every positive and negative PPI example was represented in 48 different forms (see Figure 2 and Methods). For every training set, we generated a complementary testing set that did not share the same positive PPI pair included in the training set, but included the same protein family (PFAM) domains (proteins in the same PFAM share ≥30% sequence identity) and negative PPI set (see Methods). The test set contained 4819 positive PPI instances (87.4%) and 692 negative ones (12.6%), hence, a naive predictor would predict 99.6% of all instances as positive PPI, rendering an error ≥12%. Thus, the testing set shared ≥30% sequence similarity with the training set, representing many cases of PPI. The files used for training and testing are available at https://github.com/gdelrioifc/PPI-RCC.

We first investigated whether positive and negative sets were separated by a simple distance criterion. To this end, we calculated the diameter *D* of the negative set of PPI (maximum Euclidian distance between any pair of RCCs within the same set) and compared this with the smallest distance *d* between the positive and negative sets. If positive and negative sets were separable by a distance criterion then, then *d* < *D*. Figure 3 shows the *D* and *d* values obtained for a subset (28) of the 48 training sets used in this work. We observed that these sets were not separable by a distance criterion.

Even though the sets were not separated by distance, a hyperplane was able to separate these two sets (see Methods) using RCCs built at 6 Å, hence, we concluded that these training sets were separable and consequently learnable (data not shown). To further explore these results, we observed that the distance values (*d*) for sets built at 3, 10, and 15 Å were equal to zero (see Figure 3), indicating that the positive and negative PPI were not separable at these distances used to build RCCs. The largest *d* values were observed at 6, 7, 8, and 9 Å, indicating that at such distances RCC representation of PPI facilitated the classification of positive from negative PPI. We provide a graphical interpretation regarding how the PPI space would look based on these results in Figure 4. Since we do not know the complete set of PPI, we expect that ML methods may fail to classify some PPI outside of the known regions for positive and negative. This is an intrinsic limitation of ML methods, but the geometrical representation by RCCs may provide ways to explore the limits of these positive and negative PPI regions, as discussed later.

We searched for the best ML model and corresponding parameters for each of these representations through the optimization algorithm implemented in AutoWeka (see Methods). Our first approximation included all positive and all negative instances for all training sets. The motivation was to test if the RCCs could represent PPI with an unbalanced training set; not every unbalanced training set would render a biased classification. The test set included 4819 positive PPI that did not include the same pairs of PPI in the training set but shared sequence similarity. For this test, we used the same negative set of PPI for training and testing. To evaluate the quality of the models and any potential bias in their classification, we compared the percentage of correctly classified instances (% CCI) for every model derived from the training sets against the difference of % CCI in the training set versus the % CCI in the testing set (see Methods). We observed that building RCCs with sidechains at a distance of 6 Å rendered the best classification (100% of correctly classified instances in the testing set) either adding or concatenating individual RCCs of the participating proteins. The next best models were derived by the concatenation of RCCs built at 5, 8, and 9 Å, including sidechains (see Appendix A for models hyper parameters; the actual models can be found at https://github.com/gdelrioifc/PPI-RCC). It was noticeable that the best models performed a selection of features and no RCC position was discarded, indicating that all RCC values were relevant for the classification. Furthermore, the RCCs built at these distances were among those that separated the positive from the negative PPI (see Figure 3). For this set, the best algorithm was AdaBoost with J48 (a decision tree). AdaBoost is a metaclassifier that uses a set of classifiers (in this case several J48 classifiers) to obtain a new classifier that considers the weighted predictions of each individual classifier. These metaclassifiers are particularly useful when the border between classes is not easy to detect using an individual classifier. However, the second and third best algorithms, rendering 99.9% of correctly classified instances in the testing set, were the k-nearest neighbor and locally weighted learning (LWL) algorithms, indicating that the border was not difficult to identify. On the contrary, these results indicated that the border was learnable and many ML algorithms were able to identify it; indeed, 33 different models were able to classify positive from negative PPI, with ≥87.4% of correctly classified instances in the testing set. The nature of machine learning algorithms is to report a result even when it is not the best one. Hence, it is noteworthy that the best models rendered >95% of correctly classified instances.

Since no RCC value was filtered out in the best models (those built at 5, 6, and 8 Å), we analyzed the frequency of RCC features present in any protein structure (i.e., how many times RCC1, RCC2, RCC3, …, RCC26 were found in all proteins). As shown in Figure 5, the RCCs built in the range from 7 to 8 Å with and without sidechains either adding or concatenating the RCCs from each protein had the largest proportion of nonzero values; very close to these were the RCCs built at 6 or 9 Å. It was noticeable that, of all the proteins analyzed, only RCCs built with 7 Å distance and without sidechains presented feature 16 (RCC16) with a value of zero value.

To evaluate if the positive and negative PPI displayed different distributions of RCC values, we performed a Wilcoxon test (See Methods). The comparison was performed for each of the 26 or 52 values for every protein pair included in the positive and negative PPI. As shown in Figure 6, all but one RCC value (RCC6) were significantly different in these two sets. This result was not consistent with the automatic selection of RCC features performed by AutoWeka, where no RCC coordinates/features were removed. It was expected that RCC6 and RCC16 would not be relevant for PPI classification, yet the performances of some of the trained models were perfect, suggesting that eliminating these two RCC positions may improve the classification in some of the worst models.

As noted above, the biased representation of positive examples of PPI in our trial may have induced a bias in the classification of the over-represented set (positive set). To further explore this idea, we performed a classification of PPI using diverse sampling procedures on the training sets. Each numeric representation was normalized, standardized. or kept according to its original values (see Figure 7 and Methods).

We used the conditions that rendered the lowest number of zero values on RCC positions. These included RCCs built using 7 and 8 Å; we also included the 6 Å for comparison with the conditions that rendered the best model using the full training set. For these RCCs, we scanned for the best model for all training sets via optimization performed by AutoWeka (see Methods). We generated 360 training sets that corresponded to the samplings performed to balance the training sets (see Methods). Each of the 360 models generated from the training sets were evaluated with 24 test sets (see Methods), rendering a total of 360 evaluation scores. To identify the best models, we plotted the percentages of correctly classified instances in every testing set against the differences of correctly classified instances between the testing and the training sets (see Figure 8), as we did before for the whole training set. The best five models were trained with RCCs without standardization or normalization, all using the locally weighted learning (LWL) algorithm. These models rendered almost perfect predictions, with 99.6% of correctly classified instances, supporting the results previously observed using the whole training set. This indicated that the learning rate was not due to bias in the training set, but to the separation observed between the training and testing sets. Thus, avoiding the bias composition of positive instances in the training sets rendered models that classified correctly almost all instances in the testing set. In all these models, no RCC attributes were discarded during the optimization executed by AutoWeka. The hyperparameters for all 360 models are reported in Appendix A. The actual models can be found at https://github.com/gdelrioifc/PPI-RCC.

The results of 6 Å did not improve the results obtained using 7 or 8 Å (data not shown). Comparing these results (99.6% of correctly classified instances) with those obtained with the full training set (100% of correctly classified instances in the testing set) indicated that the sampling did not reduce the positive and negative PPI regions. The fact that the LWL algorithm rendered the best predictions implied that the density of positive or negative PPI favored the corresponding class at any given point in the RCC space, even when the sampling produced the same number of positive and negative PPI or more negative than positive PPI; in other words, positive and negative PPI were separated in the RCC space (see Figure 4). LWL works in a similar way to k-nearest neighbors, but with a kernel (kernel regression); in this algorithm, a test instance of PPI was classified according to its k-nearest neighbors from the training set and a weighted contribution of each neighbor [24].

Finally, we prepared a set of positive and negative instances of PPI without redundancy, that is, no RCCs used for training were found in the testing set and no RCCs in the training or testing sets were repeated (see Methods). This required reducing the number of instances. Both training and testing sets included 1:1, 2:1, and 3:1 sets (positives:negatives), as described in Figure 7 for RCCs built at 7 or 8 Å with or without sidechains. We searched for the best models using AutoWeka, as previously described (see Methods), and the results are summarized in Figure 9. The best model was identified used the LWL algorithm based on RCCs built at a distance of 7 Å with sidechains, achieving 96% CCI in the test set. The hyperparameters for these models are available in Appendix A. The actual training and test sets can be found at https://github.com/gdelrioifc/PPI-RCC.

## 3. Discussion

Our results indicated that the packing of proteins seemed to follow a distinct pattern that was more informative when the distance criterion used to create the contact map of residues was either 7 or 8 Å apart, either including or not including sidechains. We previously reported that these same distance cut-off values rendered the best models to classify protein structure (classification based on CATH database) and protein function (Gene Ontology annotations) from RCCs [25], presumably because, at these distances, the number of RCC values close to zero is minimal.

Another important aspect of our results was the efficiency achieved in the classification of PPI; either with the full set of positive and negative PPI or by different samplings, we achieved almost perfect classification and prediction. Our analysis regarding the distance of separation between positive and negative instances of PPI indicated that unsupervised learning may be efficient in classifying PPI, and our results provided evidence that supervised learning is very efficient in classifying positive and negative PPI. It was shown that the current methods to predict PPI displayed similar accuracy to high-throughput experimental bioassays [26,27] and, in the best case, prediction methods complement the experimental results by prioritizing PPI for further experimental validation [28,29]. Thus, RCC performance is as good as or better than previous results, although this comparison should be observed with caution since different datasets were used.

For many PPI prediction methods, datasets were shown to be biased either because (i) the negative PPI dataset was not experimentally confirmed but assumed based on cellular localization [30], or (ii) the sequence similarity between training and testing sets was not considered, leading to overestimation of the prediction efficiency [31,32]. In the present work, we avoided these sources of bias in the construction of the training (Negatome database includes only experimentally validated negative PPI; see Methods) and testing datasets (no RCCs present in the training set were included in the test set). Our training and testing sets did share some degree of sequence similarity, but the separability between these sets indicated that we did not overestimate the learning rate of the best models. Considering that most proteins share some sequence similarity, we argue that our testing set is applicable for most cases. It would be relevant to evaluate RCC models to classify special cases of proteins with no sequence similarity, such as orphan proteins [33]. A more recent bias was noted involving proteins detectable by current experimental approaches, by cellular localization, or evolutionary lineage [34,35,36]; unfortunately, at this point we cannot address this bias and future studies should serve to correct or improve upon the results presented here.

We previously noted that the size (total number of amino acid residues) of proteins is linearly related to the number of clusters in the RCC [21], consistent with experimental observations that proteins present a near-constant density [37]. Based on our current results indicating that adding RCCs effectively represent PPI, it seems that protein–protein complexes may also linearly grow in number of clusters according to the number of residues (the sum of RCCs linearly increases the number of clusters with the molecular weight of the protein complex), or simply keep a constant density. In fact, it was reported that proteins and protein complexes show constant density as measured by electrospray ionization mass spectrometry and ion mobility [38]. Thus, our observation that the sum of individual RCCs serves to separate positive PPI from negative PPI may provide a geometrical approximation to study PPI that deserves to be further studied. In a similar fashion, the concatenation sets of RCCs projected two points in a 26-dimensional space into a single point of 52 dimensions; our results showed that the points in these 52 dimensions that represent positive PPI follow a distinct pattern different from those of negative PPI. This pattern may obey geometrical rules that could be studied using graph theory and/or algebra.

RCC requires knowledge of the 3D structure of the participating proteins, thereby limiting the applicability of this tool to proteins of known 3D structure. However, there are many methods to predict the 3D structure of proteins with different efficiencies [39,40]; it would be relevant to estimate how close a model should be to the real 3D structure to be useful in PPI predictions based on RCCs. The models reported here are not meant to quantify the strength of a PPI, but we anticipate this may be possible using RCCs. Our results indicated that the backbone conformation represented as RCCs contains enough information to model PPI, where the backbone conformation is the consequence of a particular set of sidechains, allowing RCCs to capture these details. Not including sidechain atoms is convenient to accelerate RCC computation, but does not make the prediction independent of the sidechains. These are the first results showing a clear relationship between protein 3D structure and PPI and highlight some intriguing possibilities that require future evaluation. For instance, it is possible that protein variants sharing very similar RCCs with the wild type keep the same interactions as the wild type sequence; in consequence, protein mutants significantly altering wild type RCCs should also alter protein interactions.

In summary, we provide the first evidence that PPI can be effectively modeled by combining individual RCCs of participating proteins. The use of RCCs provides a new perspective to geometrically study protein–protein interactions.

## 4. Materials and Methods

### 4.1. Datasets

Pairs of interacting proteins were obtained from the 3DID database [22] and the noninteracting protein pairs were derived from the Negatome database [23]. For every one of the proteins in these two sets, the corresponding three-dimensional structure was obtained from the Protein Data Bank [41]. The RCCs for each protein were calculated as previously described [21], but we varied the distance criterion from 4 to 15 Å (4, 5, 6, 7, 8, 9, 10, 11, 12, 13, 14, and 15 Å) and either included or did not include the atoms of the sidechains. Then, the resulting RCCs for every pair of proteins (positive and negative sets of PPI) were added or concatenated to produce a single numeric representation for every protein pair, with 26 or 52 features (RCC1, RCC2, RCC3, …, RCC26, or RCC52). Finally, these sets were split into one set for training and one set for testing, with the training set including all the negative PPI. The testing set split the complete 3DID dataset to include a portion of the PFAM domains found in the 3DID dataset. Particularly, if a PFAM domain (PFAM1 for instance) was found *N* times in the 3DID database, only a portion of *N* was included in the testing set; the split was determined by the interacting proteins that contained the corresponding PFAM1 domain. In other words, if PFAM1 was found in the presence of PFAM2, and PFAM2 was found *M* times, and *N* > *M*, then PFAM1 would be found in the testing set *M* times. This way, we guaranteed that most PPI instances were included in the training set. The testing set included the complete Negatome, so training and testing sets had the same information on the negative PPI.

To study the effect of class imbalance present in the full training set, we generated a total of 360 training sets with two different sampling strategies (see below): 120 normalized sets (using Weka filter weka.filters.unsupervised.attribute.Normalize), 120 standardized sets (using Weka filter weka.filters.unsupervised.attribute.Standardize), and 120 raw sets. The 360 sets contained exactly the same 8 different combinations (7 or 8 Å; with or without sidechains; concatenation or sum of RCC), plus 3 undersamples (1:1, 2:1, 3:1) randomly generated 3 times (3 × 3), that were normalized, standardized, and kept as raw, rendering a subtotal of 216 (8 × 3 × 3 × 3) training sets. Additionally, the 1:1 undersample was oversampled twice (1:2 and 1:3) and then normalized, standardized, and kept as raw, giving an additional subtotal of 144 (8 × 3 × 2 × 3) sets. Thus, a total of 360 different training sets resulted. Two types of sampling procedures were employed from the Weka package [42], namely, undersampling and oversampling. Undersampling (supervised filter SpreadSubsample) used the larger set (positive PPI) to randomly select a sample whose size would be equal (1:1), twice (2:1), or triple (3:1) the size of the negative set. Oversampling used the Synthetic Minority Oversampling TEchnique (SMOTE) filter in Weka to generate double (1:2) or triple (1:3) instances of the negative set. All these oversamplings included one copy of the original negative set. The 1:1 sample generated by undersampling was used to generate the 1:2 and 1:3 oversampling sets. For testing, we generated 24 different sets for all 360 sets, derived from RCCs built using a contact distance of 7 or 8 Å, either using or not using the sidechains and adding or concatenating RCCs; they were then normalized, standardized, and kept as raw, rendering a total of 24 (2 × 2 × 2 × 3). In summary, there were 8 testing sets for every 120 training sets for each data adjustment type (normalized, standardized, and raw). The numbers of positive and negative instances for training and testing in these sampling processes are indicated in Table 1.

A third experiment was conducted with datasets that were selected to avoid any repetition of RCCs within the same class (positive or negative) and between classes (positive and negative). In these datasets, the negative instances represented 80% in the training sets and the other 20% were posed in the testing sets. A total of 360 training sets were generated as previously described (216 training sets and 144 testing sets), including 1:1, 1:2, 1:3, 2:1, and 3:1 samplings (see Table 2).

### 4.2. Machine Learning and Statistical Testing

The best algorithms and hyperparameters for each training set were identified through Bayesian optimization implemented in AutoWeka [43]. This Weka plugin searches for the optimum algorithm and hyperparameters given a specified time for the search; in our case, we used 1500 min. Shorter times (500 min) reproduced the same results for most cases, hence, we assumed that 1500 min identified the best models.

To test for the separability of positive and negative PPI used in the training set, we conducted a simple classification with a support vector machine using a linear kernel in Weka.

To determine if the positive and negative PPI used in this study were different, we performed a Wilcoxon test and corrected the significance of the compared RCC positions using the Bonferroni or Benjamini Hochberg approaches. This test and accompanying corrections were performed using the Python libraries Pandas [44], Scipy [45], Numpy [46], Statsmodels [47], Seaborn [48], and Matplotlib [49].

## Figures and Tables

**Figure 1 ijms-21-04787-f001:**
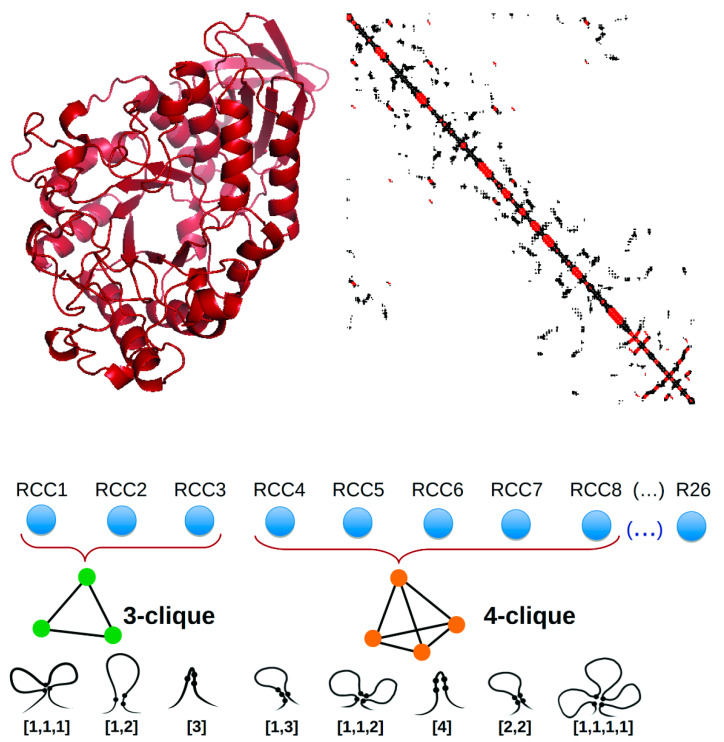
Residue cluster class (RCC) construction. Top panel: The atomic three-dimensional structure of a protein (left) is transformed into a contact map (right) that is used to build the RCC. Lower panel: The RCC is a 26-dimensional vector (RCC1, …, RCC26) derived by grouping residues close together in the three-dimensional space according to a distance criterion. These clusters group three (RCC1-RCC3), four (RCC4-RCC8), five (RCC9-RCC15), and six (RCC16-RCC26) amino acid residues; the image only represents RCC1–RCC8 for brevity. Different clusters of the same size are generated according to their sequence proximity. For instance, a cluster referred to as [1,1,1] represents three residues that are not proximal in the sequence (two or more residues apart) but are proximal to each other in the three-dimensional space.

**Figure 2 ijms-21-04787-f002:**
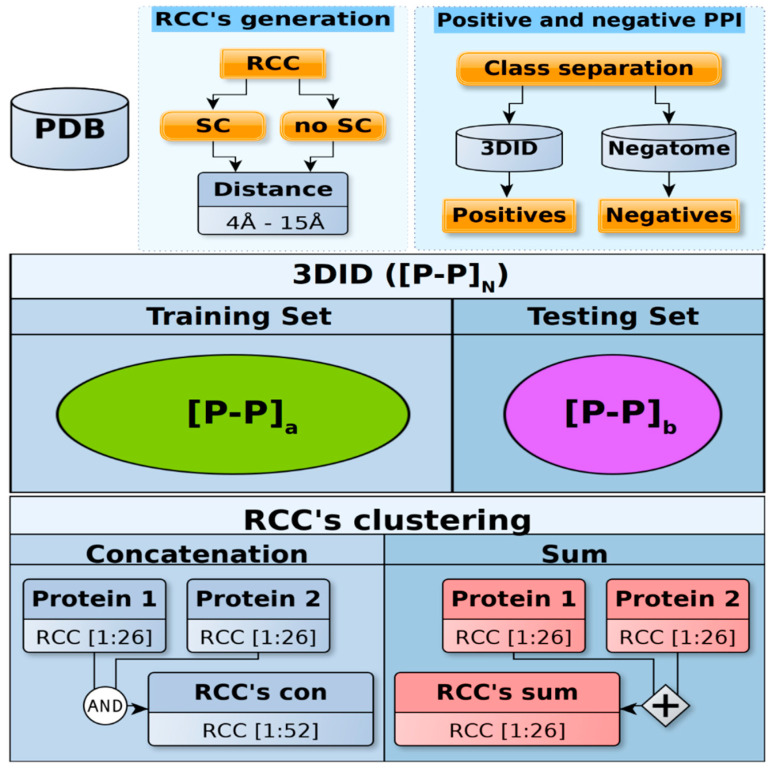
Dataset construction. Top panel: Every PDB reported as positive protein–protein interactions (PPI) in the 3DID database and as negative PPI in the Negatome database were processed to include (sidechain (SC)) or not include (no SC) the atoms of sidechains. A total of 12 different contact maps (contact distances 4–15 Å) were generated for each of these proteins and their residue cluster classes (RCCs) were calculated. Middle panel: The resulting numeric representations for all protein pairs were split into training and testing sets, which did not share the same positive PPI pair but did include the same PFAM domains. The procedure used to separate the positive PPI included in 3DID guaranteed that more instances were included in the training set than the testing set (see Methods). The positive PPI are represented in the figure as [P-P]_N_ in the 3DID, [P-P]_a_ for training, and [P-P]_b_ for testing, where *N* = *a* + *b* and *a* > *b*. Due to the limited number of negative cases, we used the same negative set for both training and testing sets. Lower panel: The sum (26 features) or concatenation (52 features) for each RCC in every pair of proteins was obtained. Finally, every PPI numeric representation was normalized, standardized, or kept in its original form (raw).

**Figure 3 ijms-21-04787-f003:**
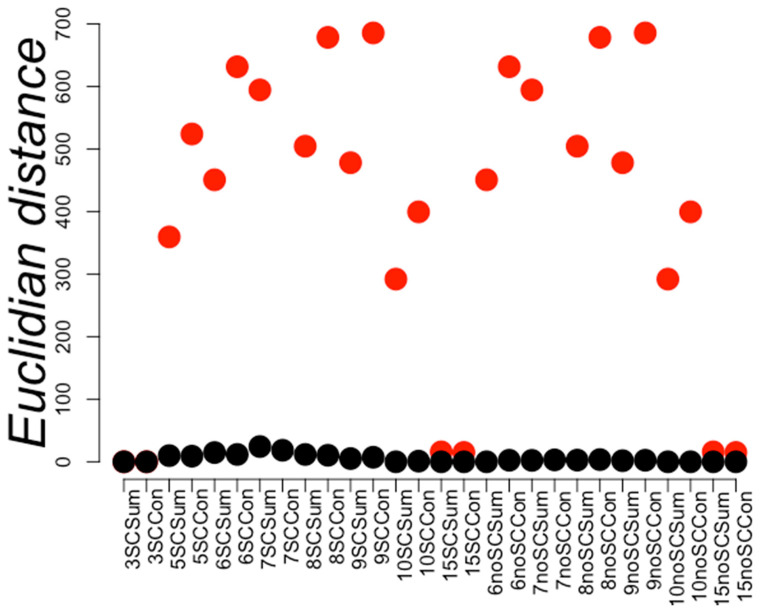
Distance within and between positive and negative PPI sets. Red circles represent the maximum Euclidian distances between instances of the same class (in this case, negative PPI) and the smallest distances between instances of different classes are shown in black circles. The *Y*-axis displays the distance values while the *X*-axis represents the different compared PPI representation compared. The first digit represents the distance used to build the RCCs and the next characters indicate whether the sidechains were included (SC) or not (noSC). Sum or Con indicates if the PPI was represented by the sum or concatenation of the individual RCC of the considered protein pair (see Methods). A red circle below the corresponding black circle indicates the separation of the positive PPI by distance from a negative PPI.

**Figure 4 ijms-21-04787-f004:**
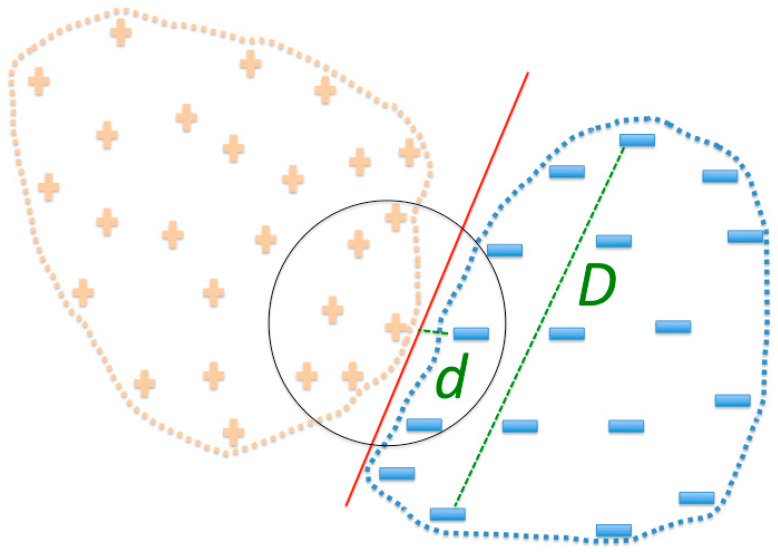
PPI space representation. Based on the distance separation of positive and negative PPI, we envisioned that positive PPI (plus symbols represent instances of positive PPI) would be dispersed over a large region in the space, but close to the negative PPI region(minus symbols represent instances of negative PPI); this proximity is likely the consequence of positive and negative PPI sharing sequence similarity. *D* represents the diameter of negative PPI, and the distance separating positive and negative PPI sets is represented by *d*. The red line separating positive and negative PPI sets represents a border that ML methods should be able to identify. The black circle represents k-nearest neighbors.

**Figure 5 ijms-21-04787-f005:**
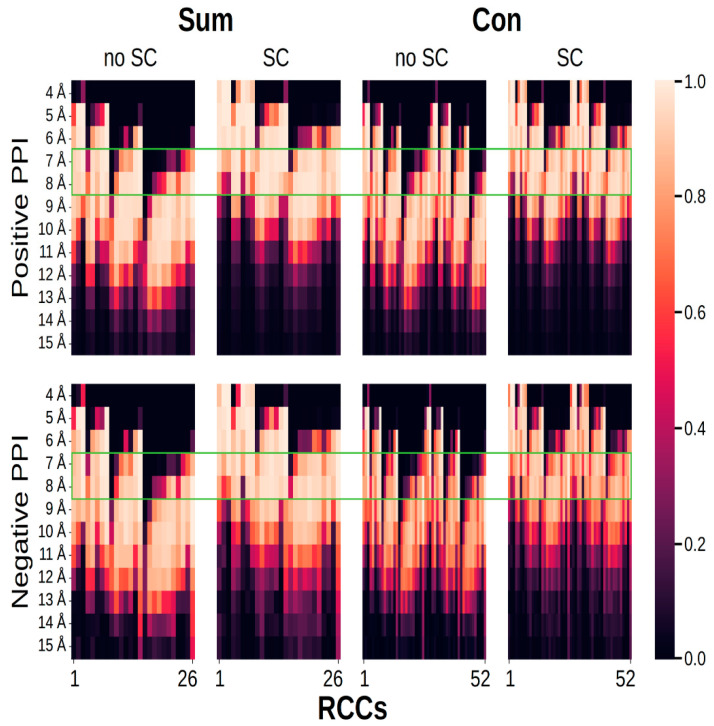
Fraction of RCCs with nonzero values. The graph shows eight different plots presenting the fraction of 26 (sum) or 52 (concatenation) values (*X*-axis) used to represent PPI (positive PPI) and pairs of proteins that did not interact (negative PPI); fractions are represented by a gradient color, with lighter colors representing those with more nonzero values. On the *Y*-axis, the distance used to build the contact map is presented (from 4 to 15 Å). The green rectangle indicates the set of distances used to build RCCs, where the number of features (RCC1, RCC2, RCC3, …, RCC26) with values equaling zero was minimum.

**Figure 6 ijms-21-04787-f006:**
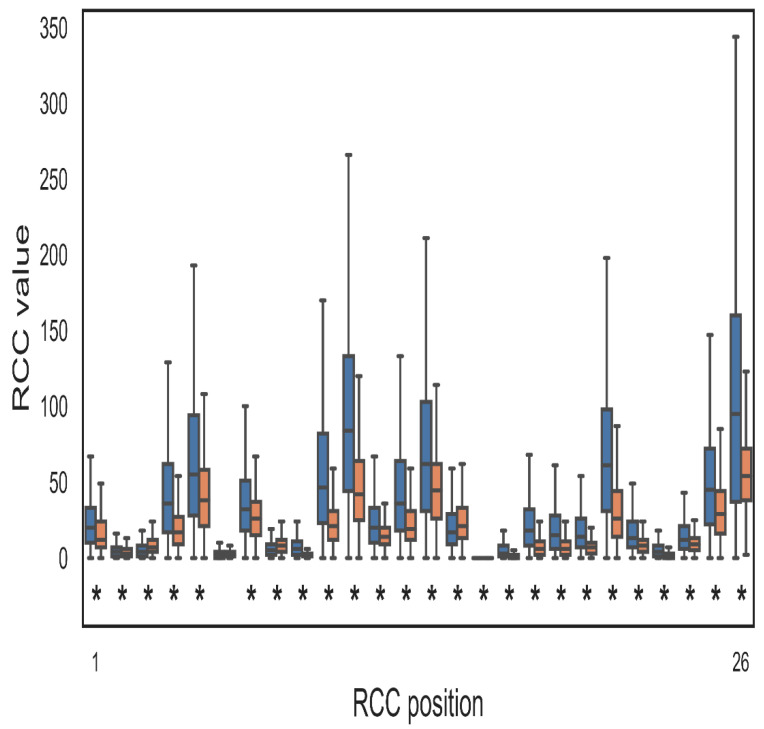
Comparison of positive and negative PPI through RCC. The image shows a representative example of the 16 performed comparisons (distance: 7, 8 Å; sidechains: yes, no; PPI RCC construction: sum, concatenation; statistical test: Wilcoxon–Bonferroni, Wilcoxon–Hochberg; all comparisons rendered very similar results (see https://github.com/gdelrioifc/PPI-RCC). The RCCs presented in the figure were obtained using a distance cutoff of 7 Å and included the residue sidechain atoms; the resulting RCCs for each protein pair were added. An asterisk is shown where the distribution of RCC values differs significantly between positive (blue) and negative (orange) PPI sets. The *X*-axis shows the RCC feature (RCC1, RCC2, …, RCC26) and the *Y*-axis represents the class of RCC and corresponding compared values. All but one RCC feature (RCC6) rendered a significantly different distribution of RCC values (*p* < 0.5; Wilcoxon test corrected by Bonferroni criterion).

**Figure 7 ijms-21-04787-f007:**
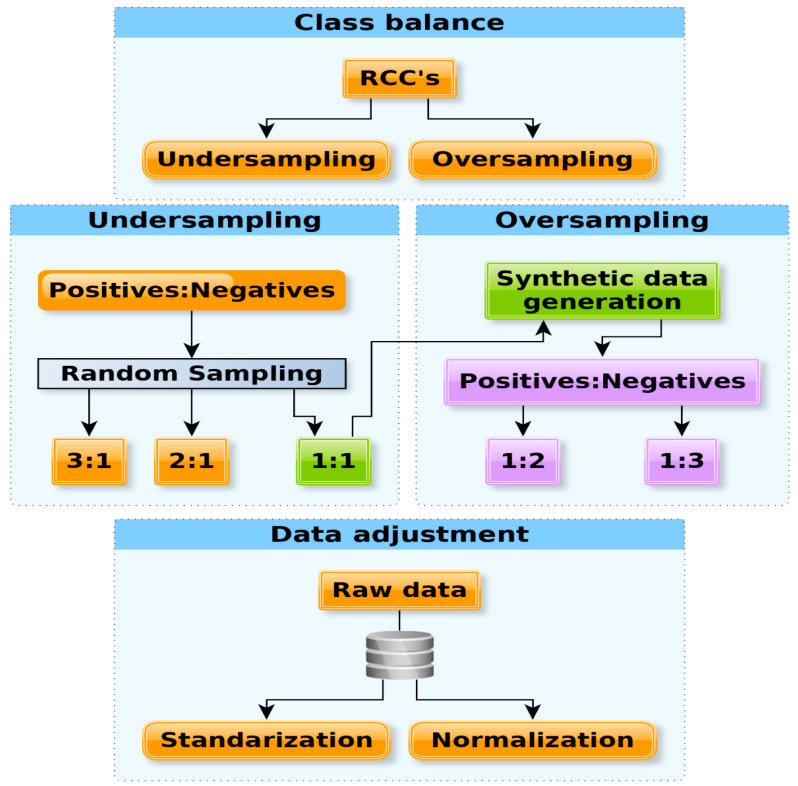
Sampling of training sets. Upper panel: Two different strategies were followed to deal with over-representation in the training sets, i.e., random elimination of instances from the over-represented positive PPI (under sampling) or generation of instances (synthetic sampling) from the under-represented negative PPI (over sampling). Middle panel: In under sampling, a random sample with equal proportions of positive and negative PPI (1:1), two times more positive than negative PPI (2:1), or three times more positive than negative PPI (3:1) were generated. In over sampling, the 1:1 sample generated in the under sampling, two times more negative than positive (1:2), and three times more negative than positive PPI (1:3) were generated. Lower panel: The RCCs generated for every PPI set were maintained (raw data), standardized, or normalized. The files used for training and testing are available at https://github.com/gdelrioifc/PPI-RCC.

**Figure 8 ijms-21-04787-f008:**
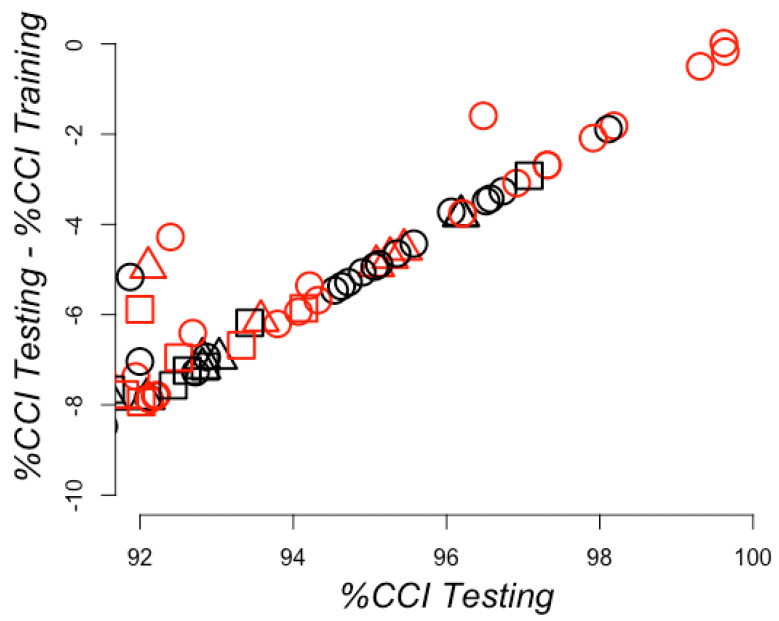
Learning efficiency on sampling training sets with redundancy. The percentages of correctly classified instances (CCI) for the testing sets (*X*-axis) are plotted against the differences observed in these values for the testing and training (*Y*-axes) sets. CCI corresponds to true predictions, including both positive and negative PPI. Therefore, 100% on the *X*-axes corresponds to not failing any prediction on the testing set and a 0 value on the *Y*-axis corresponds to no difference observed in the prediction between the training and the testing set. The best models are shown in the top right corner. Predictions achieved with raw RCCs are presented as circles, standardized RCCs as squares, and normalized RCCs as triangles. RCCs built using sidechains are otherwise shown in red and black.

**Figure 9 ijms-21-04787-f009:**
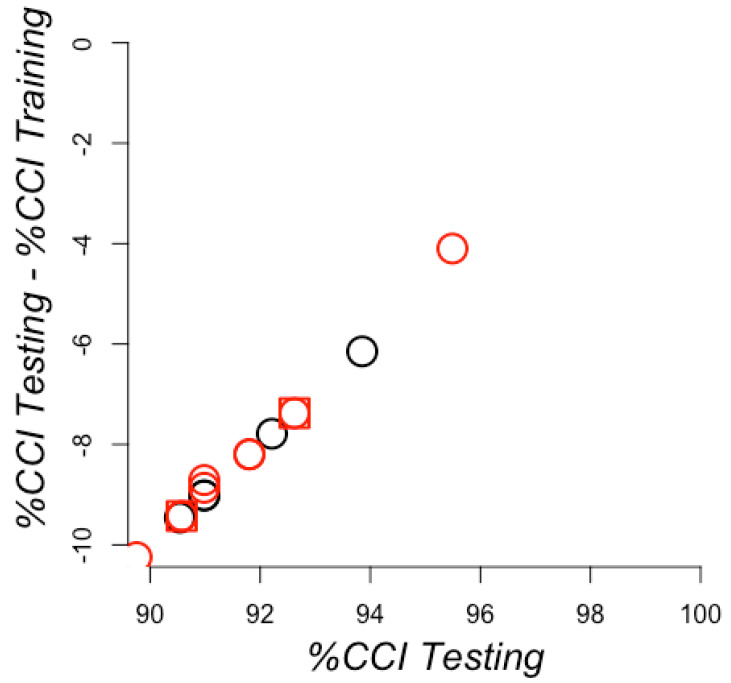
Learning efficiency on sampling training sets without redundancy. The percentage of correctly classified instances (CCI) for 360 testing sets (*X*-axis) is plotted against the difference observed in these values for testing and training (*Y*-axis). CCI corresponds to true predictions, including both positive and negative PPI. Therefore, 100% on the *X*-axes corresponds to not failing any prediction on the testing set and a 0 value on the *Y*-axis corresponds to no difference observed in the prediction between the training and the testing set. The best models are shown the top right corner. RCCs built using sidechains are otherwise represented by red circles or black squares.

**Table 1 ijms-21-04787-t001:** Test sets with redundancy.

P:N	Training		Testing	
	P	N	P	N
1:1	692	692	4819	692
2:1	1384	692	4819	692
3:1	2076	692	4819	692
1:2	692	1384	4819	692
1:3	692	2076	4819	692

P: Positive PPI; N: Negative PPI. Numbers in the table represent the number of instances for each dataset. For instance, 1:1 stands for samples with equal numbers of positive and negative instances of PPI, and 1:2 stands for samples with twice the number of negative PPI than positive PPI.

**Table 2 ijms-21-04787-t002:** Test sets without redundancy.

		Training		Testing	
		Positives	Negatives	Positives	Negatives
**Concatenation**	1:1	489	489	122	122
	2:1	978	489	122	122
	3:1	1467	489	122	122
**Sum**	1:1	448	448	111	111
	2:1	896	448	111	111
	3:1	1344	448	111	111

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
