# Peer review of "Protein–Protein Interactions Efficiently Modeled by Residue Cluster Classes"

_ijms, 2020, doi:10.3390/ijms21134787_

Round 1
Reviewer 1 Report
The authors explore the use of ML techniques which rely on clique-based residues cluster classes to classify -- learn, identify -- protein protein interactions. The research is of clear interest to the structural bioinformatics community, and of relevance to a wide range of research thursts, including drug design and the study of protein protein complexes. The authors pose a 99.6% accuracy prediction rate, which is astounding.
Major comments
The datasets used include 3DID and Negatome, which contain 171141 instances of protein protein interaction assemblies, and 692 negative ones, respectively. In other words, 99.6% of your combined data set contains entries that are labeled as PPI examples. Randomly selecting 1000 of the combined set, and randomly identifying 98% of them as PPIs, would yield approximately the success rate that you your models exhibit. How can you assure us that that is not the approach taken by your ML models?
On line 115 you mention that the training sets were separable and learnable. But doesn't that mean that you are trying to solve an easy problem, which then puts into doubt whether the 99% accuracy is noteworthy or not. If the space were not easily separable -- possibly only at some higher dimensional feature space -- then a 99% success rate would be extra worthy.
Also, please quantify what you mean by "shared some sequence similarity" on line 76. The testing and training sets should never have overlapping entries, as that would introduce overfitting and bias. Although it is true that, to some extent, all sequences have similarities because there are only 20 naturally occurring amino acids, the chance that any 10 or more contiguous amino acids are identical among any two proteins is very very small. On line 143 you also mention that you used the same negative set of PPI for both training and testing. This may be due out of necessity because the negative data set is so small, but still that's not allowed.
Minor comments
Figure 2 has an "a" instead of a "-" between 4A and 15A.
The manuscript has more than a few grammatical errors. On line 12, help improving should be help improve. On line 30, you write maintaining ... regulate, when it should be maintaining ... regulating. There are many other grammatical errors.
Author Response
- The datasets used include 3DID and Negatome, which contain 171141 instances of protein protein interaction assemblies, and 692 negative ones, respectively. In other words, 99.6% of your combined data set contains entries that are labeled as PPI examples. Randomly selecting 1000 of the combined set, and randomly identifying 98% of them as PPIs, would yield approximately the success rate that you your models exhibit. How can you assure us that that is not the approach taken by your ML models?
This is a very important point. Using a biased dataset indeed may, in some cases, lead to generate a biased model. The full training set indeed included 99.6% of positive PPI instances and the rest were negative ones; yet the test set did not contain the same proportion of positive PPI instances; our test sets include 4819 (87.4%) positive PPI and 692 (12.6%) negative ones. Thus, a naïve predictor would fail to correctly predict at least 12% of the positive PPIs, yet our results indicate that such is not the case. Furthermore, we designed a sampling procedure to eliminate this bias (see Figure 8 and Table 1). In this sampling procedure, we used training sets with PPI positive:negative ratios of 1:1, 1:2, 1:3, 2:1 and 3:1. Our results confirmed that RCC were able to correctly predict positive and negative PPI instances (>99%).
We did not clarify this aspect of the different ratios of positive and negative instances in the training and testing sets. We are now including a sentence to clarify it in lines 78-19: “the test set contained 4,819 positive PPI instances (87.4%) and 692 negative ones (12.6%), hence a naive predictor would predict 99.6% of all the instances as positive PPI rendering an error >=12%”.
- On line 115 you mention that the training sets were separable and learnable. But doesn't that mean that you are trying to solve an easy problem, which then puts into doubt whether the 99% accuracy is noteworthy or not. If the space were not easily separable -- possibly only at some higher dimensional feature space -- then a 99% success rate would be extra worthy.
Indeed, we claim that the RCC representation of PPI is learnable, and we support that claim by showing that many different ML-algorithms were able to learn PPIs with good efficiency. Machine-learning algorithms are heuristic in nature, that is, they would render a result even when this is not the best one. The striking results we found is that many ML-algorithms can actually render very high accuracies in the prediction of PPI. This is not expected from a heuristic method. The negative and positive PPI instances are not separable by a distance, but by a hyper plane, supporting the idea represented in Figure 4. We could have used a Support Vector Machine to separate every PPI representation used in our study, and in that case the inability to separate positive from negative PPI will imply the border is not trivial to learn. In either case, the results we show support the claim that PPI represented by the sum or concatenation of the corresponding RCC of the participating proteins is learnable.
The representation of PPI by RCC allows machine-learning approaches to efficiently classify positive from negative PPI; this is the result we consider noteworthy to share with the scientific community. To clarify this, we now state in lines 153-155 that “…The nature of machine-learning algorithms is to report a result even when it is not the best one. Hence, it is noteworthy that the best models rendered >95% of correctly classified instances…”.
- Also, please quantify what you mean by "shared some sequence similarity" on line 76. The testing and training sets should never have overlapping entries, as that would introduce overfitting and bias. Although it is true that, to some extent, all sequences have similarities because there are only 20 naturally occurring amino acids, the chance that any 10 or more contiguous amino acids are identical among any two proteins is very very small. On line 143 you also mention that you used the same negative set of PPI for both training and testing. This may be due out of necessity because the negative data set is so small, but still that's not allowed.
Sequences that share >= 30% of sequence identity belong to the same PFAM family. We have changed the description in lines 76-77 to clarify this: “…included the same PFAM domains (proteins in the same PFAM family share >=30% sequence identity)…”.
The use of the same negative PPI set was indeed, as the reviewer noted, to deal with the limited number of instances in this set. In addition to the controls described in the previous notes to evaluate the goodness of our models, we now further validate our claims about the efficiency of RCC to assist in classifying PPIs. We conducted both training and testing experiments avoiding repeating the same RCC on these sets; that is, no RCC used in the training set was included in the testing set and no RCC was repeated within the training or testing sets. We also used different ratios of positive and negative PPI as before: 1:1, 2:1 and 3:1. These results are presented in Figure 9 and the preparation of these sets was added to methods. This new set of results showed that the best ML-algorithm based on RCC learned with 96% of efficiency to classify PPI.
- Figure 2 has an "a" instead of a "-" between 4A and 15A.
We appreciate the note. We fixed this.
- The manuscript has more than a few grammatical errors. On line 12, help improving should be help improve. On line 30, you write maintaining ... regulate, when it should be maintaining ...regulating. There are many other grammatical errors.
We have re-organized the work considering the comments from this and other reviewers; in doing this, we corrected the grammar errors.
Reviewer 2 Report
This manuscript is not without interest but to me, it is not clear that it owns the merit to be published in a multidisciplinary review like IJMS. Amongst aspects for which I do not reckon publications are the general focus of the manuscript that appears highly specialized and the authors actually explore a lot (too much) the technical aspects without really providing a demonstration of the predictive power of their too. Elements like showing structural information that the platform could provide and for which it outcomes what already is available nowadays is important. The authors should avoid a presentation quite typical to pure computational manuscripts to catch the interest of a wide audience.
Some of the aspects missing, for example, is a good and understandable description of RCC at the early introduction. The idea is good but a bit of effort in presenting the idea, the limitation and some pictures of "real" cases would be great. There is also a problem in the organization. In fact, for RCC, we know more about it in the discussion than at the beginning of the work.
The result section is way too long, a lot of material could be passed to the ESI. Also the authors say "it would be relevant to estimate how close a model should be to the real 3D structure, to be useful in PPI predictions based on RCC." but at no point, I see any 3D structures or PPI. It is all about ML and statistics. It is really computationally focused.
The main comment is really about how the work is presented that is very close to the computation but little is shown on the structure.
For a IJMS manuscript the authors should rethink the manuscript.
Author Response
- This manuscript is not without interest but to me, it is not clear that it owns the merit to be published in a multidisciplinary review like IJMS. Amongst aspects for which I do not reckon publications are the general focus of the manuscript that appears highly specialized and the authors actually explore a lot (too much) the technical aspects without really providing a demonstration of the predictive power of their too. Elements like showing structural information that the platform could provide and for which it outcomes what already is available nowadays is important. The authors should avoid a presentation quite typical to pure computational manuscripts to catch the interest of a wide audience.
We appreciate the note. The work is not a review, but an original research work. We have adapted some parts of our work to allow a larger audience to follow our work, but we recognize that this work has a strong computational component.
- Some of the aspects missing, for example, is a good and understandable description of RCC at the early introduction. The idea is good but a bit of effort in presenting the idea, the limitation and some pictures of "real" cases would be great. There is also a problem in the organization. In fact, for RCC, we know more about it in the discussion than at the beginning of the work.
We appreciate the note. We have moved the description of RCC from the discussion to the introduction section.
- The result section is way too long, a lot of material could be passed to the ESI. Also the authors say "it would be relevant to estimate how close a model should be to the real 3D structure, to be useful in PPI predictions based on RCC." but at no point, I see any 3D structures or PPI. It is all about ML and statistics. It is really computationally focused. The main comment is really about how the work is presented that is very close to the computation but little is shown on the structure. For a IJMS manuscript the authors should rethink the manuscript.
Indeed, the focus of our work is computational. RCCs are scoring functions derived from the protein 3D structure. We used this representation to show that protein-protein interaction can efficiently being predicted by RCC and machine learning. We did not show any image for protein structures and PPI, but addressing this comment from the reviewer, we are now including in Figure 1 an image to clarify this aspect of our work.
Reviewer 3 Report
The authors use a residue cluster class (RCC) method to analyse protein pairs that interact vs. pairs that do not interact. The RCC is performed on the separate protein structures and the RCCs of the complex is just the sum of the RCCs. No interface of the protein pair is considered!!! The authors claim that it is possible just on the analysis of the sum of the separate RCCs to predict if proteins interact or do not interact by ML. This is an extraordinary claim that needs special proof. I have strong doubts that this is indeed possible (see below) mainly because it makes physically little sense. The result of the authors may have something to do with the fact that the number of "negative" PPI is very small (692). I suggest a few checks the authors should consider:
- Most protein-protein interactions can be disrupted by single point mutations at the interface (e.g. replacement of a hydrophobic residue by a charged residue at the interface). So with a single point mutation most PPI can shift to the group of non-interacting PPs. Very likely the RCCs of the protein pair will change very little by mutating a surface residue (most of the RCC will relate to burried clusters...). They authors may even want to check this on example cases.
- The authors consider in the group of interacting PPIs also many proteins that do not interact: For example, in the positive pairs A and B may form a pair and C and D form a pair. But if A and D and B and C are unrelated it is very unlikely that A and C or B and C or B and D or A and D will interact. Hence, the authors can count all these pairs also as negative pairs. This would extend the data base of negative pairs alot and provides a critical test because now the same RCCs as in the positive ppi set (but in different combinations) are also included in the negative ppi set.
- Another important test one could think of is to shuffle the surface residues in proteins that interact (this will destroy the interface). How does the sum of RCCs changes? These can then also go into the set of non-interacting protein pairs. It needs to be emphasized again that this paper makes a very stong claim that it is basically quite simple to predict if two proteins interact or not which stays in contrast to a large body of existing research! This claim needs extraordinary proof!
Author Response
- Most protein-protein interactions can be disrupted by single point mutations at the interface (e.g. replacement of a hydrophobic residue by a charged residue at the interface). So with a single point mutation most PPI can shift to the group of non-interacting PPs. Very likely the RCCs of the protein pair will change very little by mutating a surface residue (most of the RCC will relate to burried clusters...). They authors may even want to check this on example cases.
This is a good point. We looked for examples of proteins with known 3D structure that included the wild-type as well as the mutants that disrupted a PPI. We found a database that includes that information: https://life.bsc.es/pid/skempi2/. Unfortunately, all the protein mutants included in the database did not disrupt completely the PPI, only reduced the interaction. Since our method is not aimed to quantify, but to classify PPI, we would expect all the instances in this dataset would be predicted as positive PPI by our method. To test this, we analyzed the 288 pairs of wild-type and mutants PPI included in that database and observed that indeed all these pairs were predicted as positive PPI; we are including the Weka’s output for these predictions. We will be delighted to include any examples where the experimental 3D structure of a wild-type and a disruptive mutation is known, but we were not able to identify any. If the reviewer knows any of those cases and is willing to share it with us, we would be very happy to test this idea.
- The authors consider in the group of interacting PPIs also many proteins that do not interact: For example, in the positive pairs A and B may form a pair and C and D form a pair. But if A and D and B and C are unrelated it is very unlikely that A and C or B and C or B and D or A and D will interact. Hence, the authors can count all these pairs also as negative pairs. This would extend the data base of negative pairs alot and provides a critical test because now the same RCCs as in the positive ppi set (but in different combinations) are also included in the negative ppi set.
This point is important. We checked that the presence of the same negative PPI set in the training and the testing set did not affect the reliability of our predictions by conducting the samplings reported in Figure 7. Our results show that our best models were not biased.
To further confirm our results, we now conducted both training and testing experiments avoiding repeating the same RCC on these sets; that is, no RCC used in the training set was included in the testing set. We also used different ratios of positive and negative PPI as before: 1:1, 2:1 and 3:1. These results are presented in Figure 9 and the preparation of these sets was added to methods. This new set of results showed that the best ML-algorithm based on RCC learned with 96% of efficiency to classify PPI.
- Another important test one could think of is to shuffle the surface residues in proteins that interact (this will destroy the interface). How does the sum of RCCs changes? These can then also go into the set of non-interacting protein pairs. It needs to be emphasized again that this paper makes a very stong claim that it is basically quite simple to predict if two proteins interact or not which stays in contrast to a large body of existing research! This claim needs extraordinary proof!
We appreciate the comment. Indeed our claim is an important one considering the efficiency observed in classifying PPI. The aim of our work is to show the high efficiency achieved by representing proteins as RCC in the task of classifying PPI.
We did perform an experiment that somehow addresses this comment. Our RCCs were built using or not side-chains atoms. The best models included the side-chains, but some very efficient models were obtained without the side-chain atoms. Furthermore, the RCC does not consider the amino acid identity, it only takes into account the contacts based on sequence proximity. We do agree that this result is surprising considering all the different efforts that have been developed in the past to model PPI. Our results indicate that the backbone conformation represented as RCCs contains enough information to model PPI; the backbone conformation is the consequence of a particular set of side-chains present in the protein. These are the first results that show this relationship between the protein 3D structure and PPI. We feel these are relevant results that deserve to be shared with the scientific community to further explore the use and meaning of RCC for modeling protein structure and function.
Indeed, further experiments could be performed to study the use of RCC to model PPI. For instance, the question that is bringing up the reviewer is: how close a 3D model of a protein should be to the native one for RCC to efficiently predict PPI? To properly address this concern, we need to perform many different experiments. For instance, we could use conformers of one or both proteins in a PPI that deviate 1, 2, 3, 4…Å from the native one, or use protein structure decoys, like the ones proposed by the reviewer and evaluate if RCC are still good in predicting PPI. All these experiments would be relevant for protein structure modelers, who could use their models to predict PPI. At this point, our interest is to show that RCC are features that allow machine-learning approaches to effectively classify PPI. We believe the special issue on protein structure analysis and prediction with statistical scoring functions is an adequate forum to disseminate this first result and hopefully promote further experiments by the community.
In this regard, it is worth mentioning that during the revision of this work, the Entropy journal accepted for publication our work describing an efficient implementation to calculate RCCs (https://www.mdpi.com/1099-4300/22/4/472). In addition to distributing the code, we also released a RCC database derived from the last PDB release. This we hope could promote further research in the area of machine learning aimed at modeling protein structure and function, such as PPI. We are now including this publication among our references, as reference 25.

Round 2
Reviewer 1 Report
I appreciate the clarification in the text and in the response from the authors. Although I believe that the training set is too skewed with positive PPI cases, (the new) Figure 8 and Table 1 lessen my worry quite a bit.
Author Response
We appreciate the comment; the training sets were not skewed by the positive cases. We realized that we could not explain properly the structure of the new test sets. We have improved the description in Methods and in the results sections. We have added a new Table (Table 2) that describes the size of the training and testing sets
Reviewer 3 Report
The authors improved the manuscript and responded to my concerns.
I am still not convinced about the claims of the study (and therefore I agree to publish it after resppnding to the request below) and I have therefore still one important point the authors should include.
1. For example, in one response the authors stated:
...Furthermore, the RCC does not consider the amino acid identity, it only takes into account the contacts based on sequence proximity....
later, it is stated that only the backbone structure of the protein partners matter (not the side chains).
So, if a reader understands this correctly it means for a certain protein-protein interaction that the structural topology of the two partners determines if they interact since the structural topologyy of the individual proteins determines the RCCs. Hence, it means I could just synthesize two proteins (sequence does not matter as long as it folds!) that fold to certain topologies. These folded structural topologies give certain RCCs and this means the two proteins will then interact, correct?
This is still very strong claim and if my above statement is correct I request that the authors include such statement (that others can check experimentally) in the Discussion or Conclusion section.
Author Response
We appreciate the note. The question the reviewer is posing is quite interesting and we would like to provide some tentative solutions.
1) Let us clarify what the RCC is. RCC correspond with the denser region of protein structures. Such packing is the result of all interactions (contacts are defined by a distance criteria) among the atoms of a protein. Yet, for protein structure classification, only the backbone is sufficient to visually determine the protein fold. The backbone conformation is the result of the packing of the atoms in the side chains. So, even when we did not include the side-chain atoms for the construction of the RCC, their role in the protein structure is already imprinted in the backbone conformation. Not including these atoms accelerates the computation of RCC though, hence the advantage of not using them.
2) Machine-learning algorithms aim at approximating the solution to a mathematical function between an input and output variables. Lets say we have two variables, X and Y, and we wish to establish how these two are related. In our case, X represents protein-protein pairs and Y is a binary variable stating whether these pair of proteins are interacting or not. Then, from the input variable X, we aim at predicting the output variable Y: Y = f(X). Since usually the set X is incomplete, we expect to have an associated error (e) in approximating the Y value from X hence Y = f(X) + e. When e -> 0, then we claim that f(X) has been correctly modeled. As we described in Figure 4, the form of this function corresponds with a hyper plane in a 26 dimensional space where the RCC lies; we still need to have many more negative examples to validate our current observation that e -> 0. If, our results hold for many more negative data, then such hyper plane truly represents the function of protein-protein pairs that distinguish positive from negative protein-protein interactions.
Considering these two aspects of our work, then if two polypeptides are folded, and we can obtain their corresponding RCCs, applying the function f(X) should determine whether these polypeptides interact. But as explained in our first point, folding depends on all atoms in the polypeptide. Including or not the side-chain atoms in the RCC does not imply that the backbone conformation is independent of the side-chain atoms. This elimination is only relevant to accelerate the computation of RCC.
To clarify this, we now stated in the Discussion “Our results indicate that the backbone conformation represented as RCCs contains enough information to model PPI; the backbone conformation is the consequence of a particular set of side chains, so RCC are capturing these details. Not including side-chain atoms is convenient for accelerating RCC computation, but does not make the prediction independent of the side-chains”.
However, it seems reasonable to imagine that in some cases two different protein sequences would present identical backbone fold; in such case, these proteins will have the same RCC. Such proteins are expected to not affect the protein interactions. It will be fantastic to test cases where two different protein sequences (e.g., protein mutants) with the same RCC will maintain the same interactions. Actually, reviewer 1 posed a very similar question that we agreed it is of great relevance for our work and in consequence, we conducted an experiment with the SKEMPIv2 database that includes protein wild type and corresponding mutants whose structures have been solved experimentally in the presence of a protein interactor; all these mutants changed the affinity of the interaction, but the complex was still formed. SKEMPIv2 includes 149 positive protein pairs, 105 of which include wild type proteins. We looked for any pair of proteins that will have the same RCC in this database to test for this idea (see attached file named skempi_v2_RCCsDistances.csv). We could not find any pair with the same RCC, but one: 1CT2 (Thr18Ile) and 1CT4 (Val181Ile). Both proteins are mutants of the PDB entry 1CT0, which also includes a mutation (Ser18Ile), but in SKEMPIv2 is considered the wild type. While our results indicate that both mutants have the same RCC, 1CT0 does not have the same RCC than 1CT2 or 1CT4. All these mutants maintained the same interaction as the wild type. However, the rest of the mutants in the SKEMPIv2 database also kept the protein-protein interactions reported for the wild-type and in all cases our model was able to predict these pairs as interacting proteins. A real test for our predictor would require mutants that prevented the interaction (i.e., negative cases). While this is an interesting possibility, the available data does not allow us to draw any conclusion regarding the possibility to predict mutants that would alter protein interactions at this time. Considering the relevance noted by the reviewers, we included a sentence in the discussion section to refer to this idea: “Not including side-chain atoms is convenient for accelerating RCC computation, but does not make the prediction independent of the side-chains. These are the first results showing a clear relationship between protein 3D structure and PPI and highlight some intriguing possibilities that require future evaluations. For instance, protein variants sharing very similar RCC than the wild type should keep the same interactions than the wild type sequence; in consequence, protein mutants that significantly alter the wild type RCC should alter as well the protein interactions…”.

Round 3
Reviewer 3 Report
The authors responded successfully to my concerns. The additional statements give a clear message.